# Nutrition and Diet Apps: Brazilian Panorama before and during the COVID-19 Pandemic

**DOI:** 10.3390/nu15163606

**Published:** 2023-08-17

**Authors:** Sueny Andrade Batista, Alessandra Fabrino Bretas Cupertino, Ana Paula Cupertino, Raquel Braz Assunção Botelho, Juliana Pimentel, Francisco Cartujano-Barrera, Verônica Cortez Ginani

**Affiliations:** 1Department of Nutrition, University of Brasilia, Brasilia 70910-900, Brazil; acupertino1006@gmail.com (A.F.B.C.); raquelbabotelho@gmail.com (R.B.A.B.); julopespimentel@gmail.com (J.P.); vcginani@gmail.com (V.C.G.); 2Department of Surgery, University of Rochester Medical Center, Rochester, NY 14642, USA; paula_cupertino@urmc.rochester.edu (A.P.C.); francisco_cartujano@urmc.rochester.edu (F.C.-B.)

**Keywords:** nutrition, diet, content, COVID-19, app

## Abstract

In the last decade, we have seen a substantial increase in the development and use of mobile technology to improve diet and healthy eating behaviors. Objective: To describe the characteristics of nutrition and diet apps before and after the COVID-19 pandemic available in Brazil. Methods: Nutrition and diet apps were identified using the official Apple and Google stores. The search occurred in January 2020 and May 2022 in Brazil. We extracted the nutritional content and standard indicators (e.g., being developed before or after 2020, number of languages, target population, investment, prices, seller, number of reviews and downloads, consumer rating). Results: 280 apps were launched before and 411 during the COVID-19 period. Most apps were available in at least ten languages (96.6%), with no indication of age (95.6%) or partial or full cost (59%). As for the contents, 18.9% addressed personal diet suggestions; 73.4%, nutritional education; 48.8%, revenues; 35.9%, physical activity with a nutritional guide; 2.3%, nutritional recommendation for eating out; 23.9%, grocery shopping with a scan code; 32.4%, food diary; 18.9%, water intake; and 4.6%, nutrition/diseases. The data show an evolution that may have been boosted by the pandemic and that reveals a trend towards the development of apps with educational content. Conclusion: During the pandemic, there was a positive qualitative and quantitative movement in e-health regarding the promotion of education.

## 1. Introduction

The World Obesity Atlas 2022 estimates that by 2025, around 900 million adults globally will be living with obesity, and by 2030, this number can increase to a billion [1]. The Brazilian population is particularly vulnerable since the country has a high prevalence of overweight adults. The National Health Survey (2019) showed that 60.3% of adults (around 96 million people) over 18 years old are overweight, and 25.9% are obese [2]. The prevalence of adults with obesity is estimated to be 34% by 2030, with a high annual increase of 2.0%. The report also shows that, in Brazil, obesity will represent 43.7% of premature deaths from all noncommunicable diseases (NCDs). In the country, 738,371 deaths from NCDs were recorded in 2019. Of these, 41.8% occurred prematurely (between 30 and 69 years of age), resulting in a standardized mortality rate of 275.5 premature deaths per 100,000 inhabitants. In addition, they resulted in 1.8 million hospitalizations in the health system and 8.8 billion spent on hospitalizations in 2019 [3].

In this sense, smartphones and tablets can be used to manage various daily activities. Among them, there is a growing interest in the use of apps for health. In general, apps can assist in health education, disease self-management, remote monitoring, and data collection [4]. Mobile apps are also a possible tool to help with necessary nutritional changes [5]. Paramastri et al. [6], in their systematic review, described the content of mobile apps that influence the improvement of health and nutrition. Examples of this content include evaluation and improvement of food consumption, physical activity monitoring, nutritional education promotion, and weight control. Another study evaluated the effects of dietary mobile apps on nutritional outcomes in a specific group of adults with NCDs. The authors concluded that these mobile apps were helpful self-monitoring tools for achieving positive weight loss [7]. 

Kalgotra, Raja, and Sharda [8] reported a 29.9% increase in the availability of apps related to health and fitness during COVID-19. A potential explanation could be the demand for nutrition apps to manage some of the adverse results of the isolation imposed during the COVID-19 pandemic, such as reduced exercise practices and changes in eating patterns. Furthermore, weight gain was a significant concern during the pandemic, given these adjustments [9]. In their study developed during the pandemic, Zachary et al. [10] noted that 22% of the participants reported weight gain. The authors listed different eating behaviors during the isolation, such as “eating in response to sight and smell” and “eating in response to stress”. Then, the search for apps capable of helping with unhealthy habits was an option [11,12]. 

For this purpose, various nutrition apps with different contents are offered. The most common components to help change diets are innovative ways to quantify food intake and report its relationship with health aspects. The entered data can be used both for self-assessment and for sharing with dietitians, researchers, or caregivers. Cultural and environmental aspects may be associated with these records, as well as lifestyle factors such as physical activity and sleep, among other influencers on people’s diet and health [13]. 

Franco et al. (2016) [14], thinking about the importance of monitoring daily food intake to target interventions for its improvement, analyzed the main contents of popular nutritional apps. Additionally, the authors compared strategies and technologies for dietary assessment and user feedback. The study took place in the UK in November 2015 on a desktop personal computer (PC) not logged into any particular user account. The 13 selected apps revealed that they used similar methods and technologies. All apps used the food diary record and, for data input, text search and a barcode scanner. The authors noted the need for innovation to enable customization of use with technologies such as artificial intelligence, image recognition, and natural language processing.

In addition to the existing content, a study carried out by Braz e Lopes (2017) [15] in Brazil aimed to investigate the reliability of information provided by 16 apps on the Android and iOS platforms. For the evaluation, the authors compared the present information regarding specific items (a Monthly basic Brazilian food basket) with scientifically validated references. They also observed the evaluation of users use of the app. Considering that the information on the composition of the investigated foods was not fully adequate and that the app’s users evaluated it positively, the authors warned in their conclusion about the disservice provided by these instruments.

However, more studies are needed to evaluate their availability in Brazil, considering their potential to contribute to changing the prevalence of overweight, obesity, and NCDs in the population. The broad access to the internet and mobile devices allows for the popularization of mobile health apps (mHealth) for several purposes [4]. Considering that easy access to devices is a significant contributor to the use of apps, data shows that from 2019 to 2021, 84% and 90% of people had access to the internet in Brazil. In addition, in 2021, the cell phone was the main device for accessing the internet at home, being used in 99.5% of households with access to the large network [16].

Apps have a great potential to contribute by providing democratic access to health information, promoting public health, and decreasing inequalities [17]. Therefore, this study aims to describe the characteristics of nutrition and diet apps before and after the COVID-19 pandemic in Brazil. 

## 2. Materials and Methods

This study was exploratory and conducted in December 2021 and May 2022. Figure 1 describes the four phases of the study strategy. The sample was a census type; that is, the entire universe of information was consulted.

The research was carried out by seven trained researchers (authors and co-authors). After defining and presenting the contents and criteria that would indicate the presence or absence of these contents, the researchers randomly selected five apps for training. Therefore, each researcher, individually, performed the analysis of the apps and then presented their results to the group. Differences were widely discussed, with a consensus on how to use the developed protocol. All were responsible for determining the study protocol, keywords, and app search variables. The research team searched for apps related to diet and nutrition in the official Apple and Google Stores (websites and mobile devices) in Brazil. Li et al. [18] and Pagoto et al. [19] methodologies guided the group for keyword and variable definitions.

### 2.1. Nutritional Content Definition

The authors defined the contents that would be evaluated in the apps according to the recommendations of the Global Strategy on Diet, Physical Activity, and Health [20]. The document guides toward healthy eating to achieve energy balance and a healthy weight. It also defines as components of a healthy diet the reduction of salt (sodium), free sugars, and total saturated fat, excluding trans fatty acids, and increasing consumption of fruits and vegetables, legumes, whole grains, and nuts with regular physical activity. As previously mentioned, the authors randomly chose five apps and evaluated the presence of aspects identified as motivating these principles. The evaluated nutritional contents were: (1) personal diet suggestion; (2) nutritional education; (3) recipes; (4) physical activity with a nutritional guide; (5) allergies and intolerances; (6) nutrition recommendation for eating out (restaurants and similar); (7) scan code grocery shopping; (8) water intake; (9) weight loss; (10) food diary; and (11) nutrition/diseases. This means that in the description of the app, one or more of these contents was explained as the purpose of its use.

### 2.2. Inclusion and Exclusion Criteria

The keywords “nutrition” and “diet” in Portuguese were defined for the search strategy. Apps related to animal nutrition, magazines, private clinics with a first in-person appointment, professional use only (apps for dietitians, doctors, and nurses), sports use only, restaurant orders or sales, and those not directly related to food or water intake were excluded. Apps solely used to count calories and calculate body mass index (BMI) were also excluded. This content helps dietitians with their prescriptions but cannot promote changes in eating habits [21]. Calorie counting and BMI need correlation to help diet improve or change [22]. We excluded apps for the exclusive use of patients of a determined clinic (CLOSED APP), only used for physical activity (ONLY FOR PHYSICAL ACTIVITY), specific and limited for groups such as vegans, pregnant women, and nursing mothers (HEALTH SPECIFICS), and related to selling food (PRODUCTS SELLING).

### 2.3. Data Collection 

After team training, data was collected using the official Apple and Google Stores (websites and mobile devices). The platforms were chosen because they are available on the most popular operating systems worldwide (iOS and Android, respectively). Therefore, they are virtual stores that allow users to search, download, and install apps from different developers and can be accessed in several countries [23].

Each researcher looked for apps independently. Thus, we carried out the second stage of the research. Data were entered into the Excel program, and discrepancies were discussed between the evaluators and a third researcher. A fourth researcher was called when necessary until the team reached a consensus. When at least one of the experts disagreed about the content, the team opened the app together. Thus, all information that could support the final decision was sought in the description. Apps were not downloaded.

### 2.4. Other Measures

We collected the standard indicator description and general contents of each app: name, primary language and all the available languages, target population, type of store (App and/or Play Store), cost, developer, country, and year of development, the enterprise or subject responsible for each app, number of reviews, number of downloads (only available for Android apps), consumer rating if the app offers support for solving problems and doubts, number of feedbacks (to understand the coverage of each app), and the category described by the store. In the case of the variables “Consumer rating” and “Number of reviews”, due to divergent values, four values were inserted. The information was collected on mobile devices with iOS and Android systems and on the official websites of the Google Store and Apple Store. To improve the discussion, the average of these variables was calculated. The apps were separated by being developed before or after 2020 (the beginning of the COVID-19 pandemic).

### 2.5. Analyses

Data were entered and analyzed with the SPSS Statistics 29 Program. A descriptive analysis of frequency related to the searched variables and nutritional contents was performed. The mean and standard deviation were calculated to describe the number of feedbacks and downloads.

## 3. Results

The total nutrition and diet-related selected apps in the official Google Play and App Store apps (on websites and mobile devices) before and after the COVID-19 pandemic began were 280 and 411, respectively (Figure 2). Figure 2 details the number of identified and maintained apps in the study and the reasons for exclusion. The data from the countries where the apps were developed is shown in Figure 3. According to the data, the United States of America (USA) was the primary developer in both periods. Data from the apps for the two stages are detailed in Appendix A and Figure 4.

In the two analyzed periods, the most prevalent content was related to “Diet/weight management” (340 vs. 623), followed by “Advice” (247 vs. 396). Figure 4 represents the percentage of each content before and after the pandemic began, considering the total number of selected apps (280 vs. 411). The highlight was the “Nutrition education” component in both moments (73.6%; n = 206 vs. 73.2%; n = 301). It is important to emphasize that the apps present one to eight contents. Most apps, until 2019, had three contents (26.5%; n = 76), and, after 2019, the highlights were those with up to two contents (24.9%; n = 102). Only 13.8% (n = 38) and 22.9% (n = 94) presented five or more contents until and after 2019, respectively.

## 4. Discussion

The study observed an increase of approximately 46.8% (280 vs. 411) in nutrition-related apps between the phases. The SARS-CoV-2 virus was critical globally during this period, setting up the COVID-19 pandemic. The interest and activity in mHealth were significant before and during the pandemic. There has been a dynamic growth in telehealth services, which can be seen through a 25% increase in downloads of mHealth apps [24]. These digital health innovations allowed people’s access to health services during social distancing [25].

Current interest and activity around mHealth are fueled by smartphones’ popularity, universality, availability, and growing concern of individuals with their physical health [26,27]. Another pivotal point is that high mobile coverage coincides with the growth of overweight rates [28]. Therefore, mHealth technology has become important for addressing the worldwide NCD epidemic. The coexistence and relationship between obesity and the COVID-19 pandemic are already evident [29]. In the context of the COVID-19 pandemic, the healthcare industry has explored digital technologies, such as mobile apps [30], and played a positive role during this period [17].

### 4.1. Overview

As for the developing countries, the US stood out in both stages with 22.9% and 27.7%, respectively. In the first stage, Brazil ranked second with the most developed apps. However, in the second stage, it lost its position to India. This population has sought to use this technology daily. Another aspect that should be noted is that these countries suffered the most deaths from the pandemic. New ways to manage the effects of the pandemic, such as nutritional aspects, can reflect this situation [31].

The target audience’s profile was similar between the two stages. Regarding access age, most apps (96.4% vs. 95.6%) were accessible to different age groups, including children and adolescents. In this way, these individuals do not need authorization from the guardian to access them, which can become a worrying situation. For example, adolescence is a phase with significant emotional changes, frequent pessimism, and low self-esteem. For these reasons, they become susceptible to adopting inappropriate eating practices and diets to achieve a body image that they believe to be ideal [22,32]. Several studies demonstrate body dissatisfaction in this group and the easy influence of information from sources such as social media. Public authorities should give special attention to these apps and how diets are oriented [33,34].

As for the changes, the percentage of free apps has decreased (25.7% vs. 12.2%). There has been an increase of 13.4% in apps that required some payment (partial or total). At both times, most apps required financial investment (52.1% vs. 65.5%). However, paid apps require a reduced financial investment (an average value of up to 12 dollars/month) compared to a face-to-face appointment with a dietitian. For example, in Brazil, according to the National Federation of Dietitians, an individual appointment must cost at least USD 35.5 [35] (BRL 1.00 = USD 5.22) [36]. Therefore, the difference between values (195.8%) can lead customers to opt for applications.

It is unknown whether the given information has a robust scientific evidence base [37]. We observed a limited and non-individualized approach, reinforced by terms used in the description such as “diet model”, “contains more than 30 free diets”, “suggestion of menus with different amounts of calories”, “menu to lose seven kilos in a week”, among others. Li et al. [18] and Weber et al. [38] show that more efforts are needed to “develop comprehensive nutrition apps supported by scientific evidence, personalized dietary guidance, and innovative technology”.

The growing popularity of apps, plus the indication of their use by health professionals to support nutritional treatment, may justify some of the findings of this study. First, the variety of languages made available by each developer (1–10 languages; 96.6%; n = 731), as well as the non-age limitation (95.6%; n = 724), offering technical support, and expanding content types per app, may suggest an attempt to meet a demanding clientele. However, this versatility must be viewed with caution. As Braz and Lopes (2017) [15] mentioned, the quality of the information provided can be questionable. When an app is available to such a broad audience, the difficulty of customization increases.

In addition, there is an unwillingness of users to evaluate the apps or little transparency in the evaluations by the administrators of the apps. Considering all the sources of information used, a significant part did not present data on consumer rating (iOS device—40.3%; n = 305; Apple website—55.7%; n = 422; Android device—72.4%; n = 548; Play Store website—63.3%; n = 479). From another perspective, most users evaluated the apps with a score greater than 4, with the maximum ranking being 5 (iOS device—47.0%; n = 356; Apple website—42.3%; n = 320; Android device—21.0%; n = 159; Play Store website—29.5%; n = 223). A good evaluation can reflect important operational aspects, such as the practicality of use. However, it may not be the evaluation of the content conveyed by the tool, corroborating the previous discussion.

In this sense, the user’s perception of the app is fundamental. Scarry et al. (2022) [39], advancing in the analysis of apps, carried out a systematic review to investigate the impact of apps on the quality of their users’ diets. The authors selected ten studies involving 1638 participants. Most studies (n = 6) evaluated changes in users’ eating behavior. The others were aimed at reducing weight or controlling glycemia. The methodologies defined for evaluation were different in all studies. Still, they consisted of scoring the diet quality according to the purpose of the study (e.g., weight loss, sodium control, etc.) and the composition of the diet obtained with the 24 h recall. The benefit generated by using apps was evident, but long-term evaluations are essential to elucidate their impact on diet quality.

In turn, Slazus et al. (2022) [26] proposed to explore the use of apps as a tool for self-monitoring the diet and their possible role in improving food choices. By applying a questionnaire, the authors first assessed how students between 18 and 25 years old perceived the need and use of apps for dietary assessment. The answers were directed toward the desire for a healthier diet (89.7%) and weight control (74.2%). Subsequently, the authors followed 63 individuals while using an app and evaluated its usability and effectiveness. Almost all participants (93.4%) defined the app as easy to use, but with reservations about choosing food (39.3%) and determining portion sizes (63.9%). However, the use of apps facilitated the change in food intake (91.8%), improving the quality of the diet and emphasizing reducing the intake of sugary foods.

Therefore, when mobile devices are used in an assisted approach, the healthcare professional can view the results and provide feedback. They can be crucial in improving public health and reducing healthcare costs. Gains also include the ease of collecting and processing dietary data and providing new ways to quantify eating behaviors and potentially inform the diet-health relationship [13,26], thus reducing barriers to healthier eating and motivating desirable changes in behavior [40].

In the economic field, apps can serve as a lower-cost modality for adequately providing health services, with shorter hospital stays and increased capacity for health services [41,42]. Additionally, during the COVID-19 pandemic, mHealth contributed to social interaction. Face-to-face appointments with health professionals were reduced, as was the possibility of contamination [43].

Ratings attributed by users to apps depend on the type of platform. Initially, 34% of the apps received scores above four; then, only 36% did. However, most apps still need to receive this feedback (56% vs. 59%). Briggs et al. [44] showed that the average score of nutrition apps evaluated in February 2020 on both platforms was 4.6. User feedback is essential for obtaining information so that later adjustments can be made to optimize operations [45]. It can contribute to increasing the effectiveness of the results or resources of the apps—within the context of a comprehensive review—which is crucial because of the wide availability of health apps [46].

### 4.2. Contents

One of the main topics is weight loss (46.8% vs. 51.8%). As for the apps in Brazil, this content was present in more than 70% of them. The most prevalent contents in weight loss apps were nutritional education, recipes, physical activity with a nutritional guide, and a food diary—a calorie counter.

Achieving healthy weight loss is not about a “diet” or “programs”, but a lifestyle with healthy eating patterns, regular physical activity, and stress control [47]. In this sense, when analyzing the contents that are connected with “weight loss”, we noticed that the suggestion of an individualized diet (20% vs. 22.1%) and the physical activity approach (30.5% vs. 46.9%) were low in the two stages. Vasiloglou et al. [48] conducted an international survey to evaluate the perceptions and opinions of healthcare professionals concerning nutrition and diet apps. The study showed that more than half of those surveyed did not recommend nutrition and diet apps to their patients/clients, and one of the reasons for the dissatisfaction was the focus on weight loss rather than behavioral changes.

Previously, also considering the dietitians’ perspective, Chen et al. (2017) [49] investigated the use of health apps and text messages in the professional practice of dietitians in Australia, New Zealand, and England. However, the authors observed an underutilization of this tool. Despite a relatively high adherence (62%), there is no direction for changing the user’s behavior, nor is the use part of the nutritional care process.

More recently, Larson et al. (2023) [50] qualitatively studied how outpatient dietitians select and use apps to support nutrition education. According to the authors, the perception of dietitians toward the use and selection of apps encompasses four axes. The first two are related to the app itself and refer to the possibility of the tool mediating nutritional education actions for long-term behavioral changes and enabling the tracking of patient practices. The other two are related to the profession. The first is the identification of patients who will benefit from the use of the tool, and the second is the requirement for adaptation by the professional to face the barriers to patient adherence when using the app.

Generally, weight management apps are widely available, and users have increased [5], but the quality of their content has been discussed previously [51]. Despite the difficulties mentioned by the cited studies, the authors agree that there are benefits to using apps as support in therapy. The absence of content with an expanded approach that fits different realities is apparently the main barrier to the consolidation of apps in this scenario. Therefore, developing evidence-based instruments is necessary for health professionals to recommend their use, and filling these gaps should be a focus of attention for nutrition app developers.

Briggs et al. [44] tracked apps with weight management as a primary outcome from both the Apple and Google Play stores. Most nutrition apps have many contents dedicated to dietary intake, anthropometric, and physical activity tracking, but lack behavior change contents. A positive correlation was also found between the total number of contents in an app and the subscription costs. This analysis was not performed in the present paper, which suggests a further investigation of this relationship.

Other contents, such as nutritional education, can contribute to adopting a healthier diet and, consequently, weight loss. This functionality had a decrease of 5.6% (71.8% vs. 66.2%) in the app context related to weight loss. Among the contents, nutritional education was the one with the highest percentage in both periods (73.6% vs. 73.2%), as well as being stable. This result is an important advance since education should be a permanent process and a starter for self-care and autonomy. When adequate and accurate information is provided, a specific contribution is made to structural integrity, functioning, and human development [52]. Schumer, Amadi, and Joshi [53] also found in their study that this content was the most commonly reported in a previous review of diet and nutrition apps [53].

As for stress control, few apps had this content (0.7% vs. 4.1%), such as exercise for stress relief, aspects that can be associated with depression, tips to improve sleep, stress and anxiety reduction, emotional assessment, and meditation. For individuals to achieve their full responsibility, the sustained implementation of evidence-based and population-based policies is necessary, especially for the most economically vulnerable population [1,54,55].

The popularity of nutritional apps is a consensus among researchers [13,14,15,39,49,50]. The practicality offered, according to the information storage capacity of the mobile device, allows the planning of food choices and the evaluation of the meal taken at the time of the event. However, several aspects must be evaluated, such as the reliability of the data provided and the effects of its use in the medium and long term. The answers must meet the perceptions of both the professional who indicates the app and has technical knowledge for the evaluation and the lay user, who is directly exposed to the advantages and disadvantages of use. In this sense, the role of apps in supporting research and health promotion must be continually analyzed so that the benefits generated are consistent and do not lose their function [13].

### 4.3. Study Limitations

Despite the use of modern devices, the fact that new technologies are regularly available can prevent access to apps that require a greater capacity to be downloaded. Another aspect is time. The speed of launching new apps contributed to the non-reach of all apps. Therefore, it is vital to observe the temporal delimitation of the research so that it is possible to understand its capacity to cover all existing apps. Just as the facts mentioned can bias the results since the type of device as well as the time of data collection are directly related to access to the available apps, other aspects are limiting: (i) The investigation is only in iOS and Android operating systems, even though they are the most popular; (ii) The understanding of each researcher in the description of the components is another aspect, and previous training was a measure to minimize this bias; (iii) Obtaining data on the time/duration of use of these apps was not possible since (iv) the apps were not downloaded and analyzed thoroughly; (v) The increase in available apps is likely to be an expected trend, regardless of the pandemic.

## 5. Conclusions

MHealth apps related to nutrition and diet had essential growth during the COVID-19 pandemic in Brazil, a period of social restrictions and health problems. No app covers the 11 nutritional resources defined in this study, which are essential components for an adequate and planned change in an individual’s nutritional status. Most apps are aimed at weight loss, a risk factor for NCDs. It is noticed that fragmentation and the absence of adequate content can contribute to the tool’s loss of power.

There is a need to evaluate and supervise the technical and scientific quality of these apps and encourage their concomitant use with professional monitoring.

## Figures and Tables

**Figure 1 nutrients-15-03606-f001:**
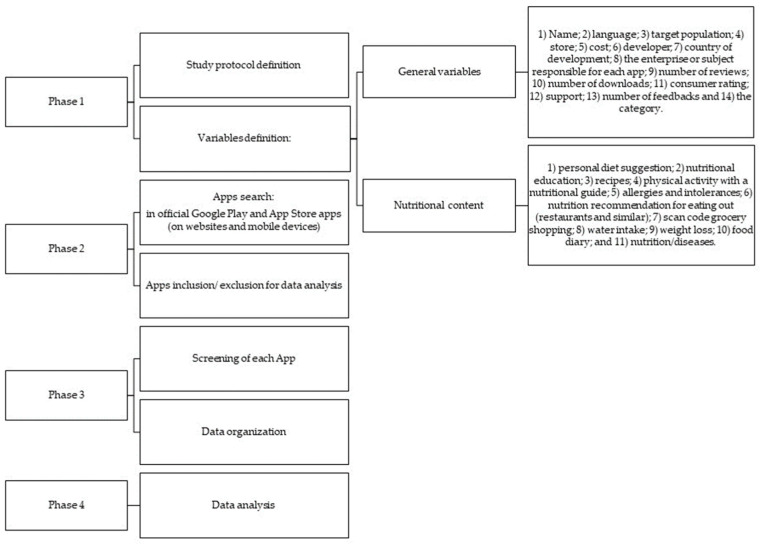
Study’s strategies to describe and evaluate Apps in nutrition.

**Figure 2 nutrients-15-03606-f002:**
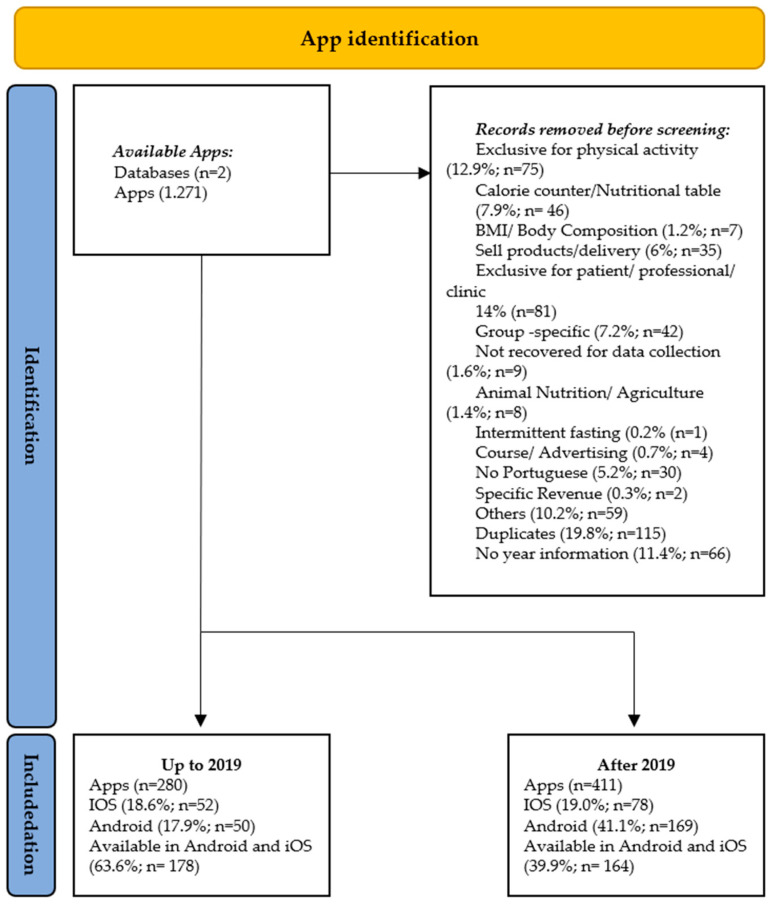
Flowchart of total nutrition and diet-related apps in official Google Play and App Store apps (on websites and mobile devices) before and after the COVID-19 pandemic.

**Figure 3 nutrients-15-03606-f003:**
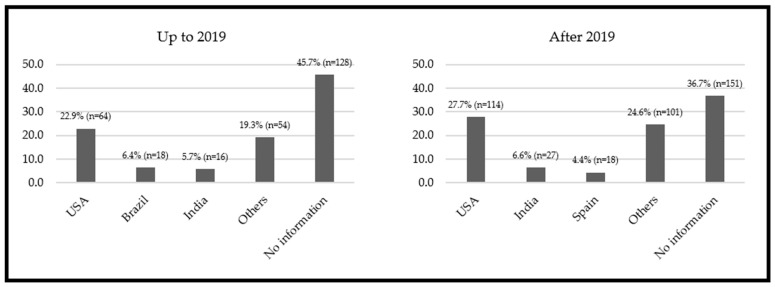
Main nutritional apps’ developer countries before and after the COVID-19 pandemic.

**Figure 4 nutrients-15-03606-f004:**
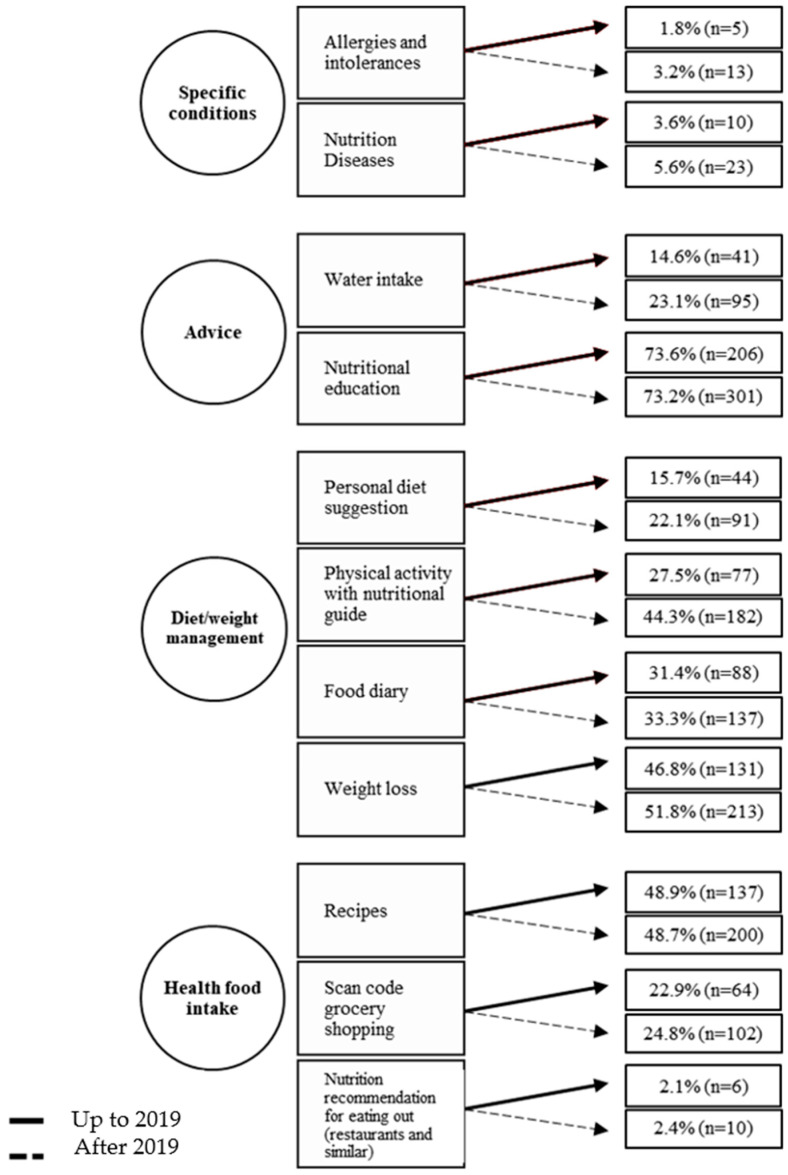
Relative and absolute frequencies of content related to nutrition and diet in apps from the official Google Play and App Store stores (on websites and mobile devices) collected from Brazil before and after the COVID-19 pandemic began.

## Data Availability

Not applicable.

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
