# Peer review of "Nutrition and Diet Apps: Brazilian Panorama before and during the COVID-19 Pandemic"

_nutrients, 2023, doi:10.3390/nu15163606_

Round 1

Reviewer 1 Report

Thank you for allowing me to review this article. The article reviews nutrition apps available in major smartphone app stores. Overall, it is an interesting piece that may prove valuable to future developers. However, a few minor aspects could use revision.

In the introduction, it would be helpful if the authors mentioned whether there are similar articles in other countries or conducted differently. Doing so would enhance the justification of their research and offer a unique perspective to their work.

Regarding the methods used, the authors mention that trained researchers conducted the research. It would be beneficial to explain the type of training they received briefly.

Some sentences are written in an excessively narrative way. For example, "The first step was to define which nutritional content would be identified in the apps. To reach this proposal, we followed the Global Strategy on Diet, Physical Activity, and Health recommendations [15]. According to the document, achieving energy balance...". I suggest the authors revise the methods and results section to avoid excessive narrative and stick to purely descriptive writing. This will not only enhance the academic quality but also make it easier to read by reducing the length of the text.

In methods, the authors note that they analyzed the Apple and Google stores. Although any reader with knowledge of these ecosystems can understand why these digital stores were chosen, the authors should argue their choice, given that, after all, it implies a selection bias that will have to be considered later.

Regarding results, it would be helpful to start by describing the total number of apps found or selected. Although this data is given in Figure 2, it would be easier if they described it in the text as their first result. In this sense, the figure takes work to follow. I recommend that the authors revise it to see if they can simplify it. The same is true in Figure 3; data that are not easy to interpret at first glance are given. The main findings could be highlighted in the text of the results. Apart from that, reading the methods section gives the feeling that more data could have been presented in tables and graphs, as more data has been collected than presented. For example, the production by country is relatively "hidden" within the flow chart of the localized apps. A graph with the countries that produce the most apps would be much more visual and easier to interpret, especially if this specific result is to be commented on in the discussion.

In discussion, for example, the authors speak of percentages of availability. However, these percentages are difficult to extract from the results. This is the case with other results discussed. They also discuss results that are difficult to locate, such as production by country, which must be searched for, unintuitively, in the flow chart of the apps found. This is the case with other results discussed in the discussion.

The limitations give the impression that limitations related to smartphone technology, app stores, or the apps themselves are discussed. However, in theory, the reader would expect to see the study's limitations reflected. For example, the selection bias, having selected two stores out of the several available, or the bias derived from the training of the researchers (which is why it is recommended that it be explained). In other words, the study has limitations that must be described so the reader can evaluate the external validity of the conclusions.

Author Response

Reviewer Report 1

Thank you for allowing me to review this article. The article reviews nutrition apps available in major smartphone app stores. Overall, it is an interesting piece that may prove valuable to future developers. However, a few minor aspects could use revision.

In the introduction, it would be helpful if the authors mentioned whether there are similar articles in other countries or conducted differently. Doing so would enhance the justification of their research and offer a unique perspective to their work.

Comment: Thank you for your comment. We attached other studies as you suggested.

Regarding the methods used, the authors mention that trained researchers conducted the research. It would be beneficial to explain the type of training they received briefly.

Comment: Thank you for your comment. We included the explanation.

Some sentences are written in an excessively narrative way. For example, "The first step was to define which nutritional content would be identified in the apps. To reach this proposal, we followed the Global Strategy on Diet, Physical Activity, and Health recommendations [15]. According to the document, achieving energy balance...". I suggest the authors revise the methods and results section to avoid excessive narrative and stick to purely descriptive writing. This will not only enhance the academic quality but also make it easier to read by reducing the length of the text.

Comment: We proofread the entire text.

In methods, the authors note that they analyzed the Apple and Google stores. Although any reader with knowledge of these ecosystems can understand why these digital stores were chosen, the authors should argue their choice, given that, after all, it implies a selection bias that will have to be considered later.

Comment: Excellent suggestion. We add this information.

Regarding results, it would be helpful to start by describing the total number of apps found or selected. Although this data is given in Figure 2, it would be easier if they described it in the text as their first result. In this sense, the figure takes work to follow. I recommend that the authors revise it to see if they can simplify it.

Comment: Thank you for your comment. We took your suggestion and added the requested explanation.

The same is true in Figure 3; data that are not easy to interpret at first glance are given. The main findings could be highlighted in the text of the results. Apart from that, reading the methods section gives the feeling that more data could have been presented in tables and graphs, as more data has been collected than presented. For example, the production by country is relatively "hidden" within the flow chart of the localized apps. A graph with the countries that produce the most apps would be much more visual and easier to interpret, especially if this specific result is to be commented on in the discussion.

Comment: Thank you for your comment. We add Figure 4 with the countries. Please, consider that we have a lot of data. So, we have decided to include part of it in the supplementary material. But if you think putting them on the text is better, we can do it.

In discussion, for example, the authors speak of percentages of availability. However, these percentages are difficult to extract from the results. This is the case with other results discussed. They also discuss results that are difficult to locate, such as production by country, which must be searched for, unintuitively, in the flow chart of the apps found. This is the case with other results discussed in the discussion.

Comment: We appreciate your suggestion and include more information in the results section.

The limitations give the impression that limitations related to smartphone technology, app stores, or the apps themselves are discussed. However, in theory, the reader would expect to see the study's limitations reflected. For example, the selection bias, having selected two stores out of the several available, or the bias derived from the training of the researchers (which is why it is recommended that it be explained). In other words, the study has limitations that must be described so the reader can evaluate the external validity of the conclusions.

Comment: Thank you for your advice.  We add the follow paragraph:

Just as the facts mentioned can bias the results, since the type of device, as well as the time of data collection, is directly related to access to the available apps, other aspects are limiting: i) The investigation is only in iOS and Android operating systems, even though they are the most popular; ii) The understanding of each researcher in the description of the components is another aspect and previous training was a measure to minimize this bias;  iii) Obtaining data on the time/duration of use of these apps was not possible since; iv) The fact that the apps were not downloaded and analyzed thoroughly; v) The increase in available apps is likely to be an expected trend, regardless of the pandemic.

Reviewer 2 Report

The submitted study aimed to identify the characteristics of nutrition/diet apps before and after the COVID-19 pandemic available in Brazil, using data from the official Apple and Google stores (i.e. data on nutritional content, year of development, number of languages, target population, investment, prices, seller, number of reviews and downloads, consumer rating, etc.). Based on the study findings 280 and 411 apps were launched before and during the COVID-19 pandemic, respectively, with most apps being available in at least ten languages (96.6%) and no indication of age (95.6%). Based on their content, 18.9% of the apps included personal diet suggestions, 73.4% nutritional education, 48.8% revenues, 35.9% physical activity with a nutritional guide, 2.3% nutritional recommendations for eating out, 23.9% grocery shopping with scan code, 32.4% food diary, 18.9% water intake, and 4.6% nutrition/diseases. According to these findings the authors conclude that during the pandemic there was a positive qualitative and quantitative progress on e-health regarding promotion of education via nutrition/diet apps.

 The authors should consider addressing the following:

1) In the objectives of the abstract, please specify that this study was specifically for Brazil, as specified in lines 70-72. In addition, the authors should present more specific details in the results that support the stated conclusion that “During the pandemic, there was a positive qualitative and quantitative movement in the e-health regard promotion of education.”.        

2) Please correct the numbering of cited refs. which starts from 25 in the introduction.

3) Please define acronyms when first used, for example define “NCDs” in line 34 of the introduction instead of line 48, BMI in line 108, EUA in figure 2, etc.

4) The “Results” section mentions Figure 1, 2 and 3, but only Figure 2 and 3 are presented in this section . This section mentions that “Figure 1 details the number of identified and maintained apps in the studies, the identification of reasons for exclusion, and data from the countries where the apps were developed”, but Figure 1 is presented before this section with its legend mentioning “Figure 1 - Study’s strategies to describe and evaluate Apps in nutrition.”. Please correct as needed.

5) Suppl. Table 1, presents the data for Target population as “<18 years 95.6% (n=724)”; “>18 years 4.2% (n=32)” and “No information 0.1% (n=1)”, these data should be better explained and discussed in the relevant sections of the manuscript.

6) The section on the limitations of this specific study should be expanded (for example, the lack of data on time/duration of use for these apps is a limitation regarding the reported conclusions of this study; the number of reviews and number of downloads for an app do not necessarily correlate to use of this app; part of the increase in available apps may be due to typical trends noted in development of apps over time rather than due to the pandemic etc.)

7) Suppl. Table 2 includes subscript "a", "b", "c" for “Number of reviews” “Number of downloads” and “Consumer rating”, respectively, but the explanations for these appear to be missing.

Proof reading of the manuscript is required to correct grammar/typographical errors, for example “Exemples...” in line 44, “Espanha” in figure 2, etc.

Author Response

Reviewer Report 2

The submitted study aimed to identify the characteristics of nutrition/diet apps before and after the COVID-19 pandemic available in Brazil, using data from the official Apple and Google stores (i.e. data on nutritional content, year of development, number of languages, target population, investment, prices, seller, number of reviews and downloads, consumer rating, etc.). Based on the study findings 280 and 411 apps were launched before and during the COVID-19 pandemic, respectively, with most apps being available in at least ten languages (96.6%) and no indication of age (95.6%). Based on their content, 18.9% of the apps included personal diet suggestions, 73.4% nutritional education, 48.8% revenues, 35.9% physical activity with a nutritional guide, 2.3% nutritional recommendations for eating out, 23.9% grocery shopping with scan code, 32.4% food diary, 18.9% water intake, and 4.6% nutrition/diseases. According to these findings the authors conclude that during the pandemic there was a positive qualitative and quantitative progress on e-health regarding promotion of education via nutrition/diet apps.

 The authors should consider addressing the following:

  • In the objectives of the abstract, please specify that this study was specifically for Brazil, as specified in lines 70-72. In addition, the authors should present more specific details in the results that support the stated conclusion that “During the pandemic, there was a positive qualitative and quantitative movement in the e-health regard promotion of education.”.        

Comment: Thank you for your suggestion. We included more information.

  • Please correct the numbering of cited refs. which starts from 25 in the introduction.

Comment: Thank you for your suggestion. We have reviewed all references.

3) Please define acronyms when first used, for example define “NCDs” in line 34 of the introduction instead of line 48, BMI in line 108, EUA in figure 2, etc.

Comment: Thank you for your suggestion. We made the changes requested.

4) The “Results” section mentions Figure 1, 2 and 3, but only Figure 2 and 3 are presented in this section . This section mentions that “Figure 1 details the number of identified and maintained apps in the studies, the identification of reasons for exclusion, and data from the countries where the apps were developed”, but Figure 1 is presented before this section with its legend mentioning “Figure 1 - Study’s strategies to describe and evaluate Apps in nutrition.”. Please correct as needed.

Comment: We have reviewed all figures and made the changes requested.

5) Suppl. Table 1, presents the data for Target population as “<18 years 95.6% (n=724)”; “>18 years 4.2% (n=32)” and “No information 0.1% (n=1)”, these data should be better explained and discussed in the relevant sections of the manuscript.

Comment: Thank you for your comment. We have reviewed all supplementary material.

6) The section on the limitations of this specific study should be expanded (for example, the lack of data on time/duration of use for these apps is a limitation regarding the reported conclusions of this study; the number of reviews and number of downloads for an app do not necessarily correlate to use of this app; part of the increase in available apps may be due to typical trends noted in development of apps over time rather than due to the pandemic etc.)

Cooment: We expand the limitation section.

7) Suppl. Table 2 includes subscript "a", "b", "c" for “Number of reviews” “Number of downloads” and “Consumer rating”, respectively, but the explanations for these appear to be missing.

Comment: Thank you for your comment. We have reviewed all supplementary material.

Round 2

Reviewer 2 Report

The authors have addressed the provided comments.